# Association between Family Environment and Adolescents’ Sexual Adaptability: Based on the Latent Profile Analysis of Personality Traits

**DOI:** 10.3390/children10020191

**Published:** 2023-01-19

**Authors:** Rui Zhao, Jun Lv, Yan Gao, Yuyan Li, Huijing Shi, Junguo Zhang, Junqing Wu, Ling Wang

**Affiliations:** 1Key Laboratory of Public Health Safety, Ministry of Education, Department of Maternal, Child and Adolescent Health, School of Public Health, Fudan University, Shanghai 200032, China; 2NHC Key Laboratory of Reproduction Regulation, Shanghai Institute for Biomedical and Pharmaceutical Technologies, Fudan University, Shanghai 200237, China; 3Key Laboratory of Health Technology Assessment, National Health Commission, China Research Center on Disability Issues, School of Public Health, Fudan University, Shanghai 200032, China; 4Division of Biostatistics, Medical College of Wisconsin, Milwaukee, WI 53226, USA; 5Department of Epidemiology, School of Public Health, Sun Yat-sen University, Guangzhou 510080, China

**Keywords:** sexual adaptability, family environment, personality, adolescents

## Abstract

Sexual adaptation plays an important role in psychosexual health. Our study aimed to investigate the relationship between the family environment and sexual adaptability among adolescents with different personality traits. A cross-sectional study was conducted in Shanghai and Shanxi province. A total of 1106 participants aged 14–19 was surveyed in 2019, including 519 boys and 587 girls. Univariate analyses and mixed regression models were performed to assess the association. Girls had a significantly lower average score of sexual self-adaptation compared to boys (4.01 ± 0.77 vs. 4.32 ± 0.64, *p* < 0.001). We found that the family environment did not impact boys’ sexual adaptation in different personality groups. For girls in a balanced group, expressiveness factors improved their sexual adaptability (*p* < 0.05), intellectual–cultural orientation and organization promoted social adaptability (*p* < 0.05) and active–recreational orientation and control decreased their social adaptability (*p* < 0.05). In the high-neuroticism group, cohesion facilitated sexual control (*p* < 0.05), while conflict and organization reduced sexual control ability, and active–recreational orientation decreased sexual adaptation (*p* < 0.05). No factors associated with the family environment were found to influence sexual adaptability in groups with low neuroticism and high ratings in other personality factors. Compared with boys, girls demonstrated lower sexual self-adaptability, and their overall sexual adaptability was more susceptible to the family environment.

## 1. Introduction

There is an increasing consensus that sexual issues are not a unique topic confined to the adult world [1]. In puberty, children develop rapidly in sexual aspects, and their accompanying mental state also changes, both of which are more likely to produce health hazards and adverse behaviors [2,3]. Psychosexual health is closely related to human sexual activities [4]. The essence of psychosexual health is a good state of individual internal sexual psychological coordination and external sexual behavior adaptation. It mainly refers to good sexual cognition, correct sexual attitude and healthy sexual behavior. Of these, sexual adaptation is an important part. In a study of Chinese adolescents, sexual adaptation meant that they could happily accept their sexual changes and consciously constrain and adjust their sexual desires and behaviors according to the requirements of social and cultural norms. Sexual adaptation includes the identification of their own gender, the adaptation to social, moral and cultural norms and regulating and controlling sexual behavior and sexual activities [5]. Studies have shown that gender disapproval and maladaptation are related to adolescent mental distress and sexual injury [6,7], and a lack of sexual control and effective sex education have been associated with risky behaviors such as adolescent dating and sexual violence [8,9], as well as an increased risk of early pregnancy [10]. Maladaptation to social norms can lead to self-doubt and self-denial, and some people cannot generally interact with the opposite sex in adulthood.

It is necessary and fundamental to investigate the current status of adolescent sexual adaptation. Several studies have found that personality, gradually formed and developed through the interaction of heredity and environment, is closely related to sexual behavior and sexual health [11,12]. Human beings, particularly adolescents, are “social animals”, and, therefore, naturally impacted by the social environments surrounding them, especially the family environment. A large number of studies have proved that there exists a significant association between the family environment and adolescent sexual development [13,14]. For example, parent–child communication is a protective factor of adolescent sexual behavior and sexual health. In addition, the more effective family monitoring is, the lower the proportion of teenagers with risky sexual behaviors [15,16].

Currently, research on sexual problems of Chinese adolescents focuses on sexual behavior, but little research has been found to study the psychosexual aspects of adolescent sexuality. In addition, few studies have addressed the critical questions regarding how the family environment affects the sexual and psychological status of adolescents, not to mention what role personality factors may play. To fill this research gap, we conducted a cross-sectional study to assess the relationship between the family environment and psychosexual health in adolescents with different personality characteristics, including the relationship with sexual adaptation.

We aim to determine whether the family environment affects sexual adaptability and whether this association is consistent among various personality traits in adolescents. Additionally, we intend to identify whether there exist different trends between the two genders previously defined regarding these associations. Therefore, more precise health interventions could be carried out in the future.

## 2. Materials and Methods

### Study Design and Participants

In November 2019, we conducted a cross-sectional study in Putuo district, Shanghai and Datong City, Shanxi province. Middle and high school students aged 12–18 were surveyed. Classes were chosen as the sampling unit, and the sampling method comprised multistage, stratified and cluster sampling. Three stages in the sampling process were as follows: (i) for each city, three middle schools, including one senior high school and two junior schools, were selected (six schools in total); (ii) for each school, 4 classes were randomly selected (24 classes in total); (iii) all the students in the selected classes were surveyed. Before the survey, we sent out passive consent letters to the parents or guardians of all subjects via their schools. The parents or guardians were required to return a signed form if they did not want their child to participate in the survey. Then, we obtained active consent from the subjects, including asking whether they agreed to participate voluntarily at the very beginning. A total of 1331 people was surveyed, 25 of whom declined to participate, leading to a response rate of 98.12%. We also conducted data quality control. For instance, 200 of the 1306 people surveyed failed to pass the lie detection questions in the later stage. In the adolescent psychosexual health scale, there are 4 pairs of polygraph questions in the design. If three pairs or more responses were inconsistent, we excluded them when analyzing the data. Thus, the final analysis ended up with 1106 subjects with a passing rate of 84.68%. As shown in Figure 1.

## 3. Measures

The *sexual adaptation subscale* from the widely accepted *adolescent psychosexual health scale* in China was employed to measure sexual adaptation (Cronbach’s α = 0.818), which included the following dimensions: sexual control (measured with six items), sexual self-adaptation (measured with five items) and sexual social adaptation (measured with nine items). The answer options of all items were “very inconsistent = 1 point”, “moderately inconsistent = 2 points”, “uncertain = 3 points”, “moderately consistent = 4 points” and “very consistent = 5 points”. Some answers were reversely scored based on the question.

The *family environment scale* (FES) [17], which is a common international scale, was employed to measure the family environment status of adolescents (Cronbach’s α = 0.826). Seven subscales with high reliability and validity in China were used in this study, including cohesion, expressiveness, conflict, intellectual–cultural orientation, active–recreational orientation, organization and a control subscale, each containing ten relevant items.

Personality was measured by using the *NEO five-factor inventory* (NEO-FFI) [18], and we used the version with 60 items (Cronbach’s α = 0.712). The scale contains five subscales: neuroticism, extroversion, openness, agreeableness and conscientiousness, each containing 12 entries.

Other demographic characteristics were measured by using a self-administered questionnaire, including demographic characteristics and knowledge, attitudes and behaviors related to psychosexual health.

## 4. Statistical Analysis

Summary tables with frequencies were provided for all the categorical variables. Chi-squared tests were used to compare the differences in demographic characteristics between sexes. For all the continuous variables, the mean and standard deviations were presented. *T*-tests were used to compare the differences in sexual control, sexual self-adaptability and sexual and social adaptation subscale scores of different sex and geographical characteristics.

Furthermore, a potential profile analysis was employed to classify adolescents with different personality traits, with a score of five dimensions on the personality scale as a manifest variable for the latent profile analysis of adolescent personality characteristics.

In the end, the mixed regression model (MRM) was used to analyze the relationship between the family environment and adolescent sexual adaptability. In addition, the latent category variables obtained from the above analysis were used as moderators to establish the mixture regression model.

All statistical tests were considered statistically significant based on the two-sided 0.05 level of significance (i.e., *p* < 0.05). All analyses were performed using SAS^®^ 9.4 and Mplus7.4.

## 5. Results

Descriptive statistics for all demographic characteristics are shown in Table 1. Among the 1106 participants aged 14–19 (SD = 2.49), 51.63% came from Shanghai and 48.37% from Shanxi. The grade levels were 58.68% in junior high school and 41.32% in senior high school. Of the participants, 82.46% reported living with their parents, 3.89% with their grandparents and 11.03% in school dormitories. In total, 86.06% reported their father’s education level to be senior high school or higher, 53.62% of whom were aged 42 years or less. Additionally, 83.54% reported their mother’s education level to be senior high school or higher, 67.73% of whom were aged 42 years or less. The per capita monthly income of 42.40% of the target families was more than CNY 6000.

The average scores for adolescent sexual adaptation are shown in Table 2. The average scores of sexual control for boys and girls were 3.54 ± 0.50 and 3.56 ± 0.44, respectively. The difference was not statistically significant (*p* = 0.329). The average social adaptation score of boys was 3.77 ± 0.68. The girls’ score for this factor was 3.76 ± 0.67. There was no statistical significance (*p* = 0.779). However, in terms of sexual self-adaptation, the boys’ average score of 4.32 ± 0.64 was significantly higher than the girls’ average score of 4.01 ± 0.77 (*p* < 0.001). Therefore, the difference between the boys’ and girls’ overall average sexual adaptation scores was significant (*p* = 0.007).

The analysis of students from different regions revealed that the average scores of students for sexual control and social adaptation were different (*p* < 0.001), and the average scores of the two factors in Shanghai students were higher than those of Shanxi students. No statistical significance was found in terms of sexual self-adaptation (*p* = 0.946).

As shown in Table 3, we selected scores on the NEO five-factor inventory as explicit variables to analyze adolescent personality characteristics based on the latent profile analysis and fitted models 1–4. The smaller the fitting index AIC (Akaike information criteria) and BIC (Bayesian information criteria) values were, the better the model fit. In this study, it was found that the AIC, BIC and aBIC decreased monotonically with increasing categories. Model entropy reached the maximum at the third classification, with 0.732 for the male model and 0.812 for the female model. Moreover, the LMR value was no longer significant in the fourth classification. Therefore, the models with three potential categories were selected as the best models.

Based on the results of the latent profile analysis, students of both male and female genders were divided into three categories according to their personality characteristics. The results of boys and girls were similar. As shown in Figure 2 and Figure 3, the first category was relatively balanced in neuroticism, extroversion, openness, agreeableness and conscientiousness, and was named the balanced group. The second category of subjects had higher neuroticism scores than the other groups, and was called the high-neuroticism group. The third category of subjects had lower neuroticism scores, but the other features were higher; we named this the low-neuroticism and high-others group. Among boys, there were 116 (22.35%) in the first category, 281 (54.14%) in the second category and 122 (23.51%) in the third category. There were 355 (60.48%) girls in the first category, 66 girls (11.24%) in the second category and 166 (28.28%) girls in the third category.

The 519 boys and 587 girls were divided into three latent categories. Possible influencing factors were included in the model, with the latent category variable “personality characteristics” as the moderator, the scores of different factors in the family environment scale as the predictor variable and the scores of adolescent sexual control, sexual self-adaptability and sexual social adaptability classified as the dependent variables. Thus, the mixed regression model of the relationship between the family environment and adolescent sexual adaptability was constructed.

The results showed that for boys with different personality traits, no family environmental factors influenced their sexual control ability, sexual self-adaptability and sexual social adaptability.As shown in Table 4, for girls in the first category, the balanced group, expressiveness factors in the family environment improved their sexual adaptability (*p* < 0.05), intellectual–cultural orientation and organization promoted their social adaptability (*p* < 0.05) and active–recreational orientation and control decreased their social adaptability (*p* < 0.05). For girls in the second category, the high-neuroticism group, cohesion in the family environment facilitated their sexual control (*p* < 0.05), while conflict and organization reduced their sexual control ability and active–recreational orientation decreased their sexual adaptation (*p* < 0.05). For girls in the third category, the low-neuroticism and high-others group, no family-environment-related factors were found to influence their sexual adaptability.

## 6. Discussion

To the best of our knowledge, there have been few studies on the relationship between the family environment and psychosexual health in China, and most studies focused on the parent–child communication [19] and parenting style [20]. Studies showing the impact of the family environment from various dimensions on the psychosexual health of adolescents are relatively rare. In addition, a large number of studies found that the formation of personality is related to the family environment [21,22], and there is a certain relationship between personality and sexual activities [23], so it is necessary to stratify personality when analyzing the impact of the family environment on adolescent psychosexual health. However, thus far, we have not found papers published on this subject. Our research results also confirmed that for adolescents with different personality traits, the influence of the family environment on their sexual adaptability was different.

Among girls, the subjects in the first group of personality traits (the group with balanced scores) had the most dimensions affected by the family environment, and a total of five dimensions affected their sexual self-adaptation and sexual social adaptation. For girls of this type, the intellectual–cultural orientation, organization, active–recreational orientation and control factors in the family environment all influenced their sexual social adaptability, among which intellectual–cultural orientation and organization had positive effects, while active–recreational orientation and control had negative effects. Family knowledge helped to develop a much healthier lifestyle, overcome difficulties and handle stress, and family intellectual–cultural orientation was the supportive basis of building a good interpersonal relationship. Studies have found that the less knowledgeable family members are, the more mental health problems teenagers have [24]. This study also showed that family knowledge of female adolescents had a positive correlation with their sexual social adaptability. Organization in the family environment is a manifestation of achieving unified action easily within a family, and girls in highly organized families in this study had better social adaptability. Adolescence is a critical period for gender identity, peer relationships and self-identity. Different teenagers of different genders express their emotions in different ways. Men often express their emotions through behavior, while women often do so through verbal communication [25]. This study showed that family expressiveness has a protective effect on female sexual self-adaptation, suggesting that positive emotional expression in the family of female adolescents is conducive to improving sexual self-adaptation.

For girls with personality traits in the second group (in which the neuroticism score was higher than the other groups), the influence of the family environment mainly focused on sexual control. Family cohesion exerts a positive influence on sexual control, and it is an important embodiment of the quality of family relationships. Other studies also show that a poor quality of family relationships is significantly related to the occurrence of teenagers browsing pornographic information and looking at pornographic pictures [26]. Conflict and organization had a negative influence on sexual control. Conflict means a disharmony of the family environment, and organization refers to a scheduled, demanding and planned mode of action in the family. Some studies show that conflict and too much emphasis on rules in the family environment cause psychological pressure on adolescents [27]. Adolescents in this family environment may be more inclined to relieve their pressure through sexual release. At the same time, research has found that active–recreational orientation can have a negative impact on teenagers’ sexual social adaptation, which may be related to the inner conflict caused by the fact that excessive emphasis on entertainment may deviate from social norms.

However, research on male students shows that the influence of the family environment on their psychosexual health is relatively low. In our study, no matter what kind of personality traits the boys had, we did not find any family environment factors that had an impact on their sexual control, sexual self-adaptability and sexual social adaptability. From the above results, we can see that, compared to the girls, the psychosexual development of male adolescents was less affected by the family environment. Other existing studies have also shown that there are gender differences between boys and girls in terms of many psychological issues [28], and that girls are more susceptible to the family environment [29]. A study on adolescents in eastern China found that the family environment would cause a higher occurrence of hysteria tendency among girls than among men. [30]. In addition, gender differences existed in other aspects of our study. We found that girls had lower sexual self-adaptability than boys, and the difference was statistically significant, which was similar to research results from other countries [31,32]. The difference is likely related to the dual perception held for both genders. Our investigation of adolescents’ sexual adaptation mainly started from the three aspects of accepting their own physical characteristics, their own gender identity and acceptance and harmony with the opposite sex. Some relevant studies have shown that a low level of self-acceptance is related to depression and anxiety [33], and also associated with low self-esteem and low life satisfaction [34]. Therefore, more attention should be paid to girls’ sexual self-adaptability, as it is of greater realistic value. Although, in recent decades, international and Chinese scholars have committed to promoting gender equality, the physiological differences between men and women, the constraints of traditional ideas and the gender division of labor and power inequality in work and family are still influential in reality. Among Chinese adolescents, there still exist gender differences in the acceptance and identification of their own gender, which suggests that a gender-differentiated intervention strategy should be adopted in the family scheme to encourage adolescents to happily accept their sexual changes and consciously constrain and adjust their sexual desires and behaviors according to the requirements of social and cultural norms.

## 7. Conclusions

We found that girls’ sexual self-adaptability was lower than boys; meanwhile, girls’ overall sexual adaptability was more susceptible to various dimensions of the family environment. For female adolescents with different personality traits, the influence of the family environment on their sexual adaptability was different. This suggests that a gender-differentiated and personality-differentiated intervention strategy should be adopted in the family scheme to promote sexual adaptation.

## 8. Limitations

Our study had some limitations. First, at present, the definition of the family environment is very broad, and Chinese and international scholars hold different opinions. Although the family environment scale (FES) is a widely used measurement tool, its application in China also has limitations (this study removed the subscale that was obviously not suitable for China), and the subscale applied in this study was not inclusive enough to cover all aspects of the family environment. Second, this study was only a cross-sectional study of two research sites, suggesting that the research results may not directly represent the overall situation of the whole country, and that there was also some uncertainty in the causal inference. In the future, more regions should be included in our longitudinal studies. Third, we found that adolescents in Shanghai and Shanxi showed differences in the scores of sexual social adaptability and sexual control, but due to the limited length of the paper, we did not further analyze the geographical differences and the possible reasons.

## 9. Future Prospects

Based on the relevant evidence provided by this study, adolescent psychosexual health interventions with different gender and personality characteristics could be conducted, which could further improve the sexual mental health level of Chinese adolescents. At the same time, we hope that the whole of society pays attention to adolescents’ psychosexual health, especially the problems related to girls’ sexual adaptability.

## Figures and Tables

**Figure 1 children-10-00191-f001:**
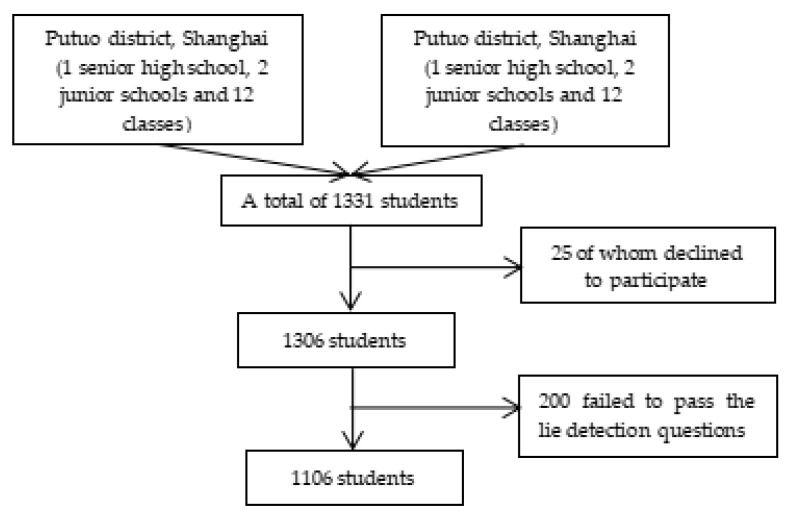
Sampling frame.

**Figure 2 children-10-00191-f002:**
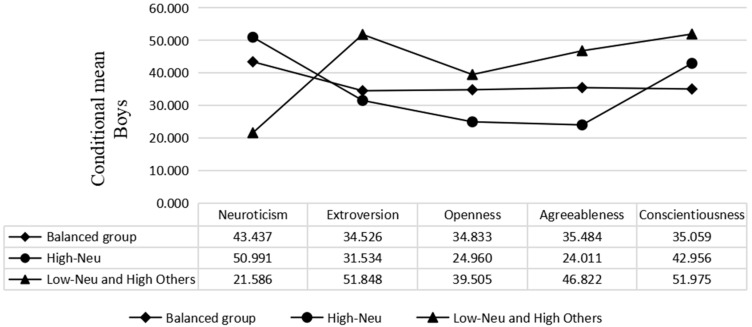
Plot of the conditional mean distribution for the latent categories (boys).

**Figure 3 children-10-00191-f003:**
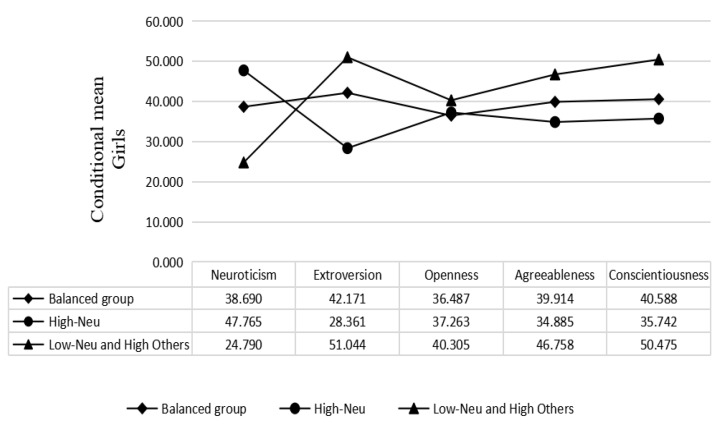
Plot of the conditional mean distribution for the latent categories (girls).

**Table 1 children-10-00191-t001:** Demographic characteristics of all respondents based on sex.

	Total (*n* = 1106)	Boys (*n* = 519)	Girls (*n* = 587)	χ^2^ *p*-Value
**Region**				0.995
Shanghai	51.63	51.64	51.62	
Shanxi	48.37	48.36	48.38	
**Grade**				0.240
Junior high school	58.68	59.54	57.92	
Senior high school	41.32	40.46	42.08	
**Main mode of residence**				0.643
Live with parents	82.46	81.50	83.30	
Live with grandparents	3.89	4.43	3.41	
Board in school	11.03	11.75	10.39	
**Father’s education attainment**				0.056
Junior high school or below	13.92	15.03	12.95	
High school, technical secondary school or vocational school	29.66	26.59	32.37	
Junior college	22.42	21.19	23.51	
Bachelor’s degree or above	34.00	37.19	31.18	
**Mother’s education attainment**				0.155
Junior high school or below	16.46	15.22	17.55	
High school, technical secondary school or vocational school	30.38	28.71	31.86	
Junior college	22.69	22.35	23.00	
Bachelor’s degree or above	30.47	33.72	27.60	
**Father’s age**				0.021
<35	1.72	2.89	0.68	
35~	51.90	52.41	51.45	
43~	35.71	35.45	35.95	
50~	10.67	9.25	11.93	
**Mother’s age**				0.381
<35	3.35	4.05	2.73	
35~	64.38	64.93	63.88	
43~	28.21	27.75	28.62	
50~	4.06	3.28	4.77	
**Per capita monthly income (CNY)**				0.544
<3000	35.08	35.07	35.09	
3000~	22.51	20.81	24.02	
6000~	20.07	20.42	19.76	
10,000~	22.33	23.70	21.12	

**Table 2 children-10-00191-t002:** Average score for each factor of sexual adaptation based on sex and region.

Subscale Score	Sex			Region		
	Boys (*n* = 519)	Girls *(n* = 587)	*p*	Shanghai (*n* = 571)	Shanxi (*n* = 535)	*p*
Sexual control	3.54 ± 0.50	3.56 ± 0.44	0.329	3.60 ± 0.48	3.50 ± 0.45	<0.001
Sexual self-adaptation	4.32 ± 0.64	4.01 ± 0.77	<0.001	4.16 ± 0.68	4.15 ± 0.78	0.946
Social adaptation	3.77 ± 0.68	3.76 ± 0.67	0.779	3.84 ± 0.70	3.68 ± 0.64	<0.001
Overall average score	3.84 ± 0.46	3.76 ± 0.46	0.007	3.85 ± 0.47	3.74 ± 0.45	<0.001

**Table 3 children-10-00191-t003:** Model comparison of latent profile analysis (personality variables).

No. of Classes	AIC	BIC	aBIC	Entropy	LMR	BLRT	The Most Likely Number of Class Members
**Boys**					-	-	
1C	17,878.351	17,920.870	17,889.128	1.000			519
2C	17,392.947	17,460.977	17,410.190	0.721	0.000	0.000	241/278
3C	17,264.324	17,357.866	17,288.033	0.732	0.003	0.000	116/281/122
4C	17,235.073	17,354.126	17,265.248	0.732	0.325	0.000	148/25/234/103
**Girls**							
1C	20,118.511	20,162.261	20,130.515	1.000	-	-	587
2C	19,557.426	19,627.427	19,576.632	0.785	0.000	0.000	392/195
3C	19,393.933	19,490.183	19,420.341	0.812	0.004	0.000	355/66/166
4C	19,355.279	19,477.780	19,388.890	0.797	0.107	0.000	17/162/84/324

**Note:** aBIC is BIC for sample correction; entropy is classification accuracy index, the value range is 0–1, the closer the Entropy is to 1, the more accurate the model classification is; LMR is likelihood ratio test index; BLRT is likelihood based on bootstrap ratio test.

**Table 4 children-10-00191-t004:** Mixture regression model of family environment and adolescent’s sexual adaptability (personality latent category variables as the moderator).

Sexual Adaptation	Family Environment Variables (Based on Personality)	*Estimate*	*S.E.*	*Est./S.E*	*p*
**Sexual Control**					
	**Girl-High-Neu Group**				
	Cohesion	0.116	0.040	2.900	0.004
	Conflict	−0.112	0.041	−2.759	0.006
	Organization	−0.100	0.051	−1.967	0.049
**Sexual Self-Adaptation**					
	**Girl-Balanced Group**				
	Expressiveness	0.108	0.025	4.425	0.000
**Social Adaptation**					
	**Girl-Balanced Group**				
	Intellectual–cultural orientation	0.043	0.020	2.088	0.037
	Active–recreational orientation	−0.048	0.020	−2.433	0.015
	Organization	0.055	0.027	2.062	0.039
	Control	−0.052	0.022	−2.386	0.017
	**Girl-High-Neu Group**				
	Active–recreational orientation	−0.270	0.132	−2.052	0.040

**Note**: *The model controls the demographic characteristics and only displays the significance variables * (*p* < 0.05).

## Data Availability

The datasets used and/or analyzed during the current study are available from the corresponding author on reasonable request.

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
