# Peer review of "Association between Family Environment and Adolescents’ Sexual Adaptability: Based on the Latent Profile Analysis of Personality Traits"

_children, 2023, doi:10.3390/children10020191_

Round 1

Reviewer 1 Report

Thank you for the opportunity to review this manuscript.

Author Response

Thank you very much for taking your precious time to review my manuscript, and your comments are very valuable to me.

Reviewer 2 Report

Dear Authors,

Thank you for this interesting piece of work, it contributes to the extant body of knowledge regarding the subject of adolescent sexuality. 

Strengths of the study

1. The data is solid and provides a substantial foundation for the assertion of the authors.

2. The Discussion section shows a thorough analysis by the authors and provides a succinct stand while maintaining academic objectivity. 

3. This reviewer is particularly impressed with the assertion of the authors that while gender equality is becoming more popular both in academia and the "real" world, there are gender differences that must be recognized and considered for effective application in psychological practice. 

Overall, the content of the paper is solid and the authors should take pride in their work. 

I do have some suggestions for the authors. Please see these suggestions below:

Major suggestions

1. In the methodology section, the authors first mention three schools selected in the first stage and then six are mentioned as being selected when the second stage is elaborated on. This is confusing. May I please suggest to the authors to clarify this. Also, perhaps the authors would insert a figure explaining the whole of the sampling process as it is difficult to understand based on the written text alone. Thank you. 

2. Kindly explain how the lie detector questions were used in the research. Perhaps it is also best if the authors mention how ethics were maintained when the lie detector questions were applied for use in this study. Thank you.

3. Could the authors please explain why they used the Adolescent Psychosexual Health Scale that is commonly used in China as opposed to a more internationally recognized scale? I do understand that the study was conducted in China, but please explain why an international scale would not have been feasible for use. This question becomes more important when the FES (an international scale) is used for measuring family environment. 

4. It would possibly be a good idea for the authors to infer from the data the reasons why the psychosexual development of male adolescents is less affected by the family environment. This inference could be placed in the Discussion section. Thank you.

Minor suggestions

1. May I please suggest that the authors not use "etc." in their manuscript? The use of it lends a certain uncertainty to the assertion of the authors. Thank you. 

2. Please check for lack of spacing between words, i.e. 14.19(SD=2.49). There should be a spacing before the bracket is opened. Thank you.

3. Page 6 line 191. Please, instead of saying "students of different genders" please say "students of both male and female genders". 

4. Page 9 lines 279-281. This sentence is odd. Please consider rewriting to capture the meaning that the authors wish to convey. Thank you. 

Language, grammar and referencing

1. Page 2 line 60. What is "adversity damage"? This is incorrect use of language. Better to change the entire section to "health hazards and adverse behaviors". Thank you. 

2. Page 2 line 86. May I suggest a change to "....few studies on the psychosexual aspects of adolescent sexuality". Thank you. 

3. Page 3 line 98. Please check the spelling for "senior". Thank you.

4. Page 3 line 109. Please change to "which includes the following dimensions". 

5. Page 4 line 154. "were showed" is incorrect, please change to "are shown". Thank you. 

6. Page 4 Table 1. Please correct the spelling for "Mother". It is incorrectly spelled "Mather" in the table. Thank you. 

7. Page 6 line 199. Please add the article "the" in front of first, second and third.

8. Page 8 line 234. "the psychosexual health" is more correct than "psychosexual health". 

9. Page 9 line 304. Please change to "and scholars at home". Thank you.

10. Page 9 lines 308-309. Instead of saying "two centers", it is more accurate to state "two research sites". Thank you.

Please check your style of referencing. It is odd that some of the references start with ".". Are these typos or have the authors inadvertently left out the names of other authors of the referenced work?

Reviewer 3 Report

I encourage to authors to remake the manuscript according the review report.

Thank you very much.

Round 2

Reviewer 1 Report

The authors have greatly improved the manuscript, please make further minor corrections.

1. The authors have added new authors, what justifies it?
2. Each table should be marked to what level of significance the authors refer, in this situation it is difficult to interpret the results.
3. Please provide information on what practical implications come from this study?

Reviewer 3 Report

Thank you

Author Response

(The authors gave the same response as above.)
